# Choosing Strategies to Deal with Artifactual EEG Data in Children with Cognitive Impairment

**DOI:** 10.3390/e23081030

**Published:** 2021-08-11

**Authors:** Ana Tost, Carolina Migliorelli, Alejandro Bachiller, Inés Medina-Rivera, Sergio Romero, Ángeles García-Cazorla, Miguel A. Mañanas

**Affiliations:** 1Biomedical Engineering Research Centre (CREB), Department of Automatic Control (ESAII), Universitat Politècnica de Catalunya (UPC), 08028 Barcelona, Spain; carolina.migliorelli@upc.edu (C.M.); alejandro.bachiller@upc.edu (A.B.); sergio.romero-lafuente@upc.edu (S.R.); miguel.angel.mananas@upc.edu (M.A.M.); 2CIBER de Bioingeniería, Biomateriales y Nanomedicina (CIBER-BBN), 28029 Madrid, Spain; 3Institut de Recerca Sant Joan de Déu, 08950 Barcelona, Spain; ifmedina@sjdhospitalbarcelona.org (I.M.-R.); agarcia@sjdhospitalbarcelona.org (Á.G.-C.); 4Neurometabolic Unit and Synaptic Metabolism Lab, Neurology Department, Institut Pediàtric de Recerca, Hospital Sant Joan de Déu, metabERN and CIBERER-ISCIII, 08950 Barcelona, Spain

**Keywords:** Rett Syndrome (RTT), electroencephalography (EEG), artifact detection, data distribution, energy function, accelerometer

## Abstract

Rett syndrome is a disease that involves acute cognitive impairment and, consequently, a complex and varied symptomatology. This study evaluates the EEG signals of twenty-nine patients and classify them according to the level of movement artifact. The main goal is to achieve an artifact rejection strategy that performs well in all signals, regardless of the artifact level. Two different methods have been studied: one based on the data distribution and the other based on the energy function, with entropy as its main component. The method based on the data distribution shows poor performance with signals containing high amplitude outliers. On the contrary, the method based on the energy function is more robust to outliers. As it does not depend on the data distribution, it is not affected by artifactual events. A double rejection strategy has been chosen, first on a motion signal (accelerometer or EEG low-pass filtered between 1 and 10 Hz) and then on the EEG signal. The results showed a higher performance when working combining both artifact rejection methods. The energy-based method, to isolate motion artifacts, and the data-distribution-based method, to eliminate the remaining lower amplitude artifacts were used. In conclusion, a new method that proves to be robust for all types of signals is designed.

## 1. Introduction

Electroencephalogram (EEG) has been consolidated over the years as one of the main techniques to identify brain activity and behavior [1]. The measurement of neurophysiological changes related to postsynaptic activity in the neocortex is a powerful tool for the study of complex neuropsychiatric disorders, since variations in EEG brain rhythms can depict a definite type of brain abnormality [2]. Neural information is extracted by designing and developing signal processing algorithms in order to use it for diagnosis, monitoring, and treatment of the identified brain pathologies. It is fundamental to make sure that the signal that will be analyzed corresponds only to the brain activity, and therefore, results will be reliable. This is incompatible with the presence of artifacts, which lead to an obstacle to interpret EEG signals due to their high amplitudes [3].

The challenge comes when signals to be dealt with are from awake pediatrics patients with a severe neurodegenerative disease [4]. Literature covers all known artifact reduction methods but, mostly, under controlled circumstances [5]. For example, during children’s sleeping periods, where the sensitivity to artifacts is highly reduced because movement, eye blinks, and muscle artifacts mostly disappear [5]. Constrained signals without uncontrolled big artifacts are well studied and numerous solutions are already presented [6,7,8]. But, what happens when patients present movement disorders, seizures, or autistic behavior? Or even all at the same time? This is what occurs in the case of patients with Rett Syndrome.

Rett Syndrome (RTT) is a neurodevelopmental disorder caused by a mutation in the X-linked dominant MECP2 gene. The occurrence varies between 1/10,000 and 1/15,000 new-born females that follow a normal development during the first 6 to 18 months of age [9]. After this period, patients will lose the abilities acquired until that moment and will initiate a degenerative process that is divided in four stages [10]. Our database includes patients in stages II, III, and IV. Patients between 18 and 36 months of age, which are in the second stage, are characterized by a severe autism, violent screaming crisis, bruxism, convulsive crisis, and an incipient loss of motor skills, as well as apraxia and ataxia. This stage is known as destructive or rapid regression because the symptomatology rapidly worsens, making evident the existence of the disease. The third phase of pseudo-stagnation (stage III) begins when the patient is over 3 years old. There is a regression of autistic manifestations and a partial recovery of visual communication. However, severe seizures start going on and epilepsy introduces a new impediment for the EEG recording. Moreover, due the degenerative nature of the disease, intellectual impairment becomes evident. Finally, from 6 years on, the fourth stage of late motor impairment arrives. Scoliosis worsen until being severe, introducing postural complications that make it impossible to stay still. A decrease in seizures is observed, which introduces a positive fact. Nevertheless, the appearance of cachexia, dystonia, and trophic disorders continue to complicate the EEG acquisition [9,11]. From early onset (Stage I) and throughout life, patients suffer from stereotyped movements (especially from the hands) and twitches. 

Recent studies have proposed evaluating the level of cognitive performance objectively despite the severity and variability of symptoms [12,13]. Analyzing the power of the EEG signals at the different frequency bands before, during, and after visual stimulation tasks allows to know the awareness, learning, and comprehension capacities of patients. The combination of cognitive training and EEG is a powerful tool to better understand the disease and to incorporate communication improvements in the patient’s day-to-day life.

Ensuring the quality of the data constitutes the first step of the analysis. Previous studies with children with similar pathologies have shown difficulties in obtaining an EEG without artifacts such as: patients are excluded due to excess artifacts; studies are based on signals no longer than 1 min due to the lack of artifact-free epochs; or visual inspections to manually remove artifacts are done, which is highly time consuming [14,15]. One of the most common artifacts in RTT analysis is movement. RTT patients are characterized by movement disorders such as stereotypes, which may involve the whole body and start at an early age; twitches (considered as manifestations of “brainstem immaturity [16]), which generally affect eyes, face, and head and finally, body spasms, which are mainly caused by spasticity and hyperreflexia [17]. Movement disorders as those observed in RTT are commonly found in other pathologies causing the same type of EEG artifact. Therefore, this study is not designed for RTT exclusively, it also aims to be applicable to numerous pathologies with similar symptoms. Some other syndromes that meet these conditions are: Williams, Fragile X, Landau-Kleffner, Prader-Willi, Angelman, tardive dyskinesia, etc., in addition to common forms of autism.

Knowing how movement affects the EEG and how to detect it is essential to designing the optimal strategy to eliminate it. Low-frequency harmonics are related to head movement, which in some cases causes pulling on the leads and small changes in the position of the electrodes, resulting in large artifacts on the EEG [18]. The accelerometer is an external device that is placed on the center of the nape and measures head movements in the x, y, and z dimensions, relative to the starting position and orientation of the accelerometer. Artifacts related to head movement are observed in the accelerometer signals below 10 Hz, while removing low frequency drifts in baseline below 1 Hz [18,19]. One of the most robust methods to analyze movement is by filtering the accelerometer between 1 and 10 Hz and evaluating the resulting signal, since the accelerometer is exclusively affected by the head movement. This is not the case of the EEG, which measures the brain activity while being influenced by other artifacts such as blinking, breathing, heartbeat, or muscle contractions. 

The most common method to remove artifacts from the EEG has been to rule out the activity that exceeds a certain threshold (usually ± 150 μV). An alternative to this method is based on the same idea of using a threshold but calculated from the data distribution, with the mean and standard deviation (*SD*) of each EEG channel [20]. An adaptive threshold is obtained using a *k-factor*: mean + *k-factor* × *SD*. The most commonly used *k-factors* in bibliography are 3, 4, and 5 [21]. This method works well for signals performed in a controlled situation where participants are asked to reduce as much as possible its head movement. In contrast, high amplitude outliers caused by uncontrolled movement will negatively affect the threshold, causing it to rise too high due to high mean and *SD*. This presents a problem because the rest of the lower amplitude artifacts that also need to be removed will not be detected. This includes those motion artifacts that do not directly affect the electrodes. To address this, our study introduces a new artifact rejection method to remove the effect of movement artifacts. This algorithm will not be based on the data distribution. The threshold will be calculated based on an energy function that has entropy as its main component. It achieves a greater adaptability and gains robustness when eliminating outliers. In addition, after movement correction, a second rejection step based on entropy or data distribution is proposed in order to remove the effect of other typical EEG artifacts.

As far as we know, the joint use of the method based on data distribution and the method based on entropy has not been found in previous studies. Furthermore, the entropy threshold has been used to detect events, but not for the rejection of epochs. The introduction of this novelty aims to achieve a compromise of valid epochs in all contexts, from the lowest to the highest level of movement artifact. It should be noted that the objective is to offer an effective solution for the clinical environment that does not require expert supervision nor high computational cost. Therefore, despite being a new methodology, it seeks to maintain the compromise between simplicity and robustness.

## 2. Materials and Methods

### 2.1. Participants, Data Acquisition, and Pre-Processing

The database is formed by 29 patients with Rett Syndrome recruited from the pediatric neuroscience unit at the Sant Joan de Déu Hospital, Barcelona. EEG signals were continuously recorded using a 20-channel EEG system (Starstim 20 wireless device from Neuroelectrics, Barcelona, Spain). Active electrodes were placed in accordance with the International 10–20 system (Fp1, Fp2, F7, F3, Fz, F4, F8, T7, C3, Cz, C4, T8, Cp5, Cp6, P7, P3, Pz, P4, P8 and Oz). Additionally, an accelerometer was placed at the back of the head to record the movement in x, y, and z axes. During the session, the subject used an eye-tracking device from Tobii Technology to record its visual response to a given stimulus on the screen. A total of 84 records, containing basal and activity periods, were registered referenced to the right ear with a sample frequency of 500 Hz. 

The study was approved by the local ethics committee following the Declaration of Helsinki (current version: Fortaleza, Brazil, October 2013) and was carried out in accordance with the protocol and with the relevant legal requirements to carry out a clinical trial with a medical device, so the project was proposed following the requirements of RD 1591/16 October 2009, which regulates medical devices. Written informed consent were obtained from all parents or legal caregivers. 

### 2.2. Influence of Patient’s Symptomatology in Data Variability

The clinical characteristics of the subjects were important to understand the high variability of the data. There were subjects who remained fairly stable during the session with slight random movements, patients who had a constant and continuous repetitive movement, and patients who produced very strong and sudden movements at times but who remained mostly stable throughout the session. This is due to the symptoms of each patient as well as the stage of the disease in which they are. The degree of attention, understanding, motor control, autism or epilepsy varies in each child and influences their behavior during the activities. Table 1 contains a summary of the principal characteristics of the patients that were registered for this study. The clinical picture of the database evidenced the variability that was then reflected in the signals. 

### 2.3. Strategies for Classifying Signals According to the Level of Movement 

The accelerometer is the best tool for measuring head movement, since it exclusively measures the displacement of the head in the three axes (*x*, *y*, and *z*) while placed on the subject’s nape. It was low-pass filtered between 1 and 10 Hz with elliptic bandpass filter of infinite impulse response (IIR). In addition, in order to sharpen the instants of movement, the signal was derived to maximize the slope changes and had a better detection of head movements. 

An alternative to the accelerometer for detecting head movements would be to filter the EEG signal between 1 and 10 Hz [19]. It must be considered that despite being filtered it is still a band of the EEG. This means that it will be influenced by brain activity and artifacts that affect low frequencies (ex.: poor electrode contact) [22]. 

To predict the percentage of artifacts that are caused by movement, as well as knowing the degree of influence of the EEG signal, the normalized cross-correlation was calculated among the envelope (calculated using the Hilbert transformation) between the following signals: EEG (1.5–80 Hz)—EEG Motion band (1–10 Hz), EEG (1.5–80 Hz)—Accelerometer and EEG Motion band (1–10 Hz)—Accelerometer. The formula used to calculate the normalized cross-correlation was: (1)$${\hat{R}}_{xy,\;coeff}\left(m\right)=\frac{1}{\sqrt{{\hat{R}}_{xx}\left(0\right){\hat{R}}_{yy}\left(0\right)}}{\hat{R}}_{xy}\left(m\right)$$
where, ${\hat{R}}_{xy,\;coeff}\;$ expresses the result of the normalized cross-correlation. ${\hat{R}}_{xx}$ and ${\hat{R}}_{yy}$ are the autocorrelations that attain its maximum value at zero lag (perfect match) and ${\hat{R}}_{xy}$ is the cross-correlation where the index *m* is the shift parameter. The order of the subscripts, with x preceding y, indicates the direction in which one sequence is shifted relative to the other: ${\hat{R}}_{xy}\left(m\right)$ = ${\hat{R}}_{yx}\left(-m\right)$.

#### Data Classification Based on the Data Distribution

The high symptomatic variability in RTT patients induced a high variability on the EEG data. Therefore, it was interesting to classify the EEG registries according to their movement patterns (described in Section 2.2) based on the mean and *SD* of the EEG filtered between 1 and 10 Hz. Finding the relationship between these variables was the key point for ranking. It was expected to find a group with low mean and *SD*, due to the lack of sudden movements and high stability; another group with high mean but low *SD*, due to constant repetitive movements and, finally, a third group with high mean and *SD,* due to sporadic strong movements. 

Two classification methods were used to analyze the group distribution of RTT patients: The Kernel Density Estimation (KDE), which represents the data using a continuous two-dimensional probability density curve that is analogous to a histogram and the Gaussian Mixture Model (GMM), which depicts the density representation as the weighted sum of Gaussian distributions. The GMM algorithm was applied to the dataset for fitting three mixture-of-Gaussian models and to assign each record to the Gaussian model it mostly belongs to [20]. As it is a probabilistic model, we filtered by probabilities to keep only those records with a probability (p) greater than 95% of belonging to its group, obtaining clearly differentiated data.

Since the accelerometer data are sensible only to head movements, the classification was performed by using the EEG data filtered between 1 and 10 Hz. This data also included other artifact sources closely related to RTT symptoms such as muscular contractions or bad electrode contact.

### 2.4. Artifact Rejection Methods 

#### 2.4.1. Signal Distribution (Mean + *SD*)

Mean and *SD* were calculated on each channel of the envelope of the EEG signal filtered between 1 and 10 Hz using the Hilbert transformation. This method requires a *k-factor* that multiplies the *SD* to fit the data correctly. The three main *k-factors* that are typically used are 3, 4, and 5. However, given the nature of our signals and its variability, more values were analyzed to ensure the optimal behavior. Its performance detecting outliers is highly dependent on the signals. If the data present outliers with great amplitude or in a great number, the method tends to increase mean and/or SD, only discarding the most prominent outliers. On the contrary, if the distribution does not present remarkable anomalies, the method may be too restrictive and discard great amounts of elements. For this reason, it has to be applied to signals with a stable and a priori known distribution.

#### 2.4.2. Signal Entropy

An alternative and robust way to detect outliers is based on an energy function that has entropy as its main component. The entropy was computed with the Shannon Entropy function, which returns the entropy for the joint distribution corresponding to object matrix C and a probability vector P [23]. Below, the whole procedure is described. The filtered signal is divided into 100-ms segments to calculate the wavelet entropy of the autocorrelation of each segment. Then, the baseline is computed as the *95%* percentile of the entropy value distribution and the threshold is determined as the *k*-value % of the obtained baseline. As the threshold is determined by the *k*-value-quantile of the absolute value of the baseline, the purpose of this algorithm is to determine the baseline of the signal without the influence of the movement peaks. This means, having the advantage of a threshold that does not depend on some events such as artifacts. Therefore, this method is more robust to outliers and more stable to signal-to noise ratio [9]. It should be noted that this method is exclusively designed to detect outliers, not being efficient to remove elements of high amplitude that are not far away from the distribution of the data. Compared with the signal distribution method, the detected threshold won’t significantly fluctuate depending on the number or the amplitude of the outliers. 

To decide the *k*-value used to calculate the threshold, five different values were tested. The analysis had been carried out observing the percentage of detected artifact-epochs against each *k*-value. The objective was to find an inflection point where the greatest change is observed and keep the *k*-value prior to that change. 

#### 2.4.3. Combined Method: Signal Distribution + Signal Entropy

We have designed a joint approach that enhances the advantages of both methods. As the band of movement or the accelerometer presented highly remarkable outliers, we expected to have a better rejection performance using the signal entropy threshold. However, once these outliers were removed, the EEG signal still presented some artifacts that could come from other sources. These artifacts showed higher amplitudes than the baseline but they were not as prominent as the former ones, so could not be considered as outliers. Thus, the signal distribution method seemed more suitable for handling this type of data.

The strategy was the same, whether the accelerometer or the band of movement was used, however, the *k-factor* for the second step varied depending on which one was used. As we expected a higher correlation for the band of movement (see Section 2.3) more artifacts presented in the original EEG signals were removed. Therefore, the resulting amplitude was lower, requiring a higher *k-factor*. If the accelerometer was used, the original EEG presented some artifacts that were not removed in the first stage because they had a different origin. For this reason, a lower *k-factor* was used. For the second step, the objective was achieving the saturation point with the best performance.

#### 2.4.4. Statistical Analysis

The different methodologies were validated by comparing the results of the automatic selection with those from the manual rejection done by two experts, considering the selection by coincidence. The manual rejection was performed with 20 registries, approximately the 23% of the total number of records. A total of sixty 5-s epochs were visually analyzed for each randomly selected record, achieving a 100-min manual rejection of signals belonging to the three classified groups. The epoch when the analysis begins was also chosen at random. 

The manual rejection was compared with the automatic rejection by calculating precision, recall, F1-score, and accuracy of each registry analyzed. The following formulas correspond to the statistical variables that were evaluated.
(2)$$Precision=\frac{TP}{TP+FP}$$
(3)$$Recall=\frac{TP}{TP+FN}$$
(4)$$F1-score=2\cdot \;\frac{Precision\cdot Recall}{Precision+Recall}$$
(5)$$Accuracy=\frac{TP+TN}{TP+TN+FP+FN}$$

An epoch was considered *positive* when it was artifact-free and *negative* when it contained an artifact. *TP* is *True Positive*, *TN* is *True Negative*, *FP* is *False Positive*, and *FN* is *False Negative*. 

Precision measures the relationship between positive epochs detected as positive (*TP*) and all epochs predicted as positive (*TP* and *FP*). Epochs marked as negative by the experts that were not detected by the automatic rejection were considered false positives. Therefore, the goal was to have the lowest FP to achieve the highest precision. On the other hand, recall measures the relationship between positive epochs detected as positive (*TP*) and all the real positive ones (*TP* + *FN*). When an epoch was marked as positive by the experts and the automatic method marked it as negative, it was considered as a false negative. In order to get a high recall, the number of false negatives had to be the lowest as possible. The balance between precision and recall is calculated with the F1-score, indicating which threshold performs the better in both variables at the same time. Finally, the accuracy measures the percentage of successful detections. The maximal accuracy is obtained by having high *TP* and *TN* and low *FP* and *FN*. 

## 3. Results

### 3.1. Data Classification

Figure 1 shows the mean and *SD* distribution of the EEG signal filtered between 1 and 10 Hz. Figure 1a shows the histogram of the mean values, where three main peaks were observed: the first one was the highest, showing that the majority of records had a low mean. Then, a second group with intermediate mean values was found; where, the greater the mean was, the lower the number of records. Finally, there was a third group of high mean, which had the lowest frequency and, therefore, showed a more limited group of records. Figure 1b shows the histogram of the *SD*. There was one main peak at low *SD* values, indicating that most records had a low *SD*. Then, a second large group was differentiated with an intermediate but still low *SD* values. Finally, a progressive increase of the *SD* appeared, until reaching a minor group of records that had a high *SD*. So far, and based on histograms, the presence of the three expected groups was observed. 

Figure 1c shows the mean-*SD* ratio distribution using KDE. Three groups were differentiated according to the density distribution: the area of highest density with low mean and low *SD*, the second highest density area with higher mean and equal or slightly higher *SD* and, finally, the smallest group with high mean and high *SD* (except for 1 record, which *SD* value was similar to the one observed in the second group). These three groups were easily differentiated in Figure 1d, which shows the results from the application of the GMM algorithm: LmLsd (Low mean Low *SD*, total: 51 records), low mean due to stable behavior during the sessions with low *SD* due to the lack of strong sudden movements. MmMsd (Mid mean Mid *SD*, total: 25 records): mid mean due to constant repetitive movement during sessions and still low or higher *SD*s, depending on whether there was the presence of stronger sporadic movements. HmHsd (High mean High *SD*, total: 8 records): both the mean and the *SD* were high because subjects presented very strong and sporadic movements, which means that they remained mainly calm during the session but the outliers caused by these movements had extremely high amplitude, which increased the global mean and the *SD*. 

### 3.2. Selection of the k-Value for the Artifact Rejection Method Based on Signal Energy

#### 3.2.1. Motion Rejection

The first step was a removal of outliers caused by motion on the accelerometer or movement band signal. In both cases, the behavior of the 5 *k*-values (from 99.95% to 99.99%) was evaluated against the percentage of epochs rejected. It was observed how the inflexion point was common for both signals, standing at 99.98%. Therefore, this *k*-value was set as the optimal for the outlier’s rejection over both motion signals.

#### 3.2.2. Amplitude Rejection

The second step was the removal of artifactual epochs that remained in the EEG signal after the removal of outliers caused by movement. In this case, the *k*-value was chosen by statistical analysis, comparing the automatic method based on entropy against the expert’s manual rejection. As the first step of outlier’s rejection was performed with two alternatives signals (accelerometer or band of movement), Figure 2 and Figure 3 show the two different statistical analysis of double energy-based rejection for both strategies. 

Figure 2 shows the boxplot representation of precision, recall, F1-score, and accuracy of the energy-based method of double rejection. The first rejection was performed with the accelerometer signal with a *k*-value of 99.98%. The second rejection was performed on the remaining epochs of the EEG signal with five different *k*-values (from 99.5% to 99.9%). Regarding precision it was observed how, as the *k*-value increased, median values decreased and dispersions increased, indicating less precision as constraint diminished. The opposite behavior was observed in the recall box plot. As the *k*-value increased, medians also increased and a dispersion decreased, meaning that the recall performance improved as restrictiveness decreased. As for F1-score it was observed how, until the third *k*-value of 99.7%, dispersion decreased as the *k*-value increased and medians remained almost stable with a minor reduction. At 99.8%, the median gently increased and dispersion remained equal. Finally, for the greatest *k*-value of 99.9%, dispersion increased and the median was drastically reduced. The last statistical variable to be analyzed is accuracy. In this case, during the first three *k*-values dispersion and median remained almost stable, achieving the highest median and lowest dispersion at 99.7%. After that, medians began to decrease and dispersions increased, indicating a deterioration in performance. 

Under a global analysis, the best performance was achieved with the third *k*-value of 99.7%. F1-score and accuracy were the ones that had the highest median together with the lowest dispersion, considering the required compromise between both factors. It had a median precision of 92.72% and a median recall of 96.23%, indicating that the artifact epochs were well selected without compromising those artifact-free ones. 

Figure 3 shows the box plots of the same statistical variables with the energy-based method of double rejection but performing the first removal on the movement band signal. Precision and recall had a behavior similar to that of Figure 2: loss of precision in exchange for a gain in recall as the *k*-value increased. Concerning F1-score and accuracy, there were two *k*-values that showed the best behavior: 99.9% and 99.92%. Median values were equal for accuracy but, in the case of F1-score, the median value was higher with a *k*-value of 99.92%. Regarding dispersion, it was higher with a *k*-value of 99.92% for the F1-score. On the contrary, in the case of accuracy, the *k*-value of 99.9% had a higher dispersion. As differences were minor, it was concluded that both *k*-values showed comparable results with a good performance. 

### 3.3. Selection of the k-Factor for the Artifact Rejection Method Based on Signal Distribution

#### 3.3.1. Artifact Rejection Method Based on Signal Distribution

The most common adaptive method in EEG signals to remove artifacts was based on the distribution of the signal, making use of its mean and *SD* multiplied by a *k-factor*. The lower the *k-factor*, the lower the threshold and the greater the restriction. Given the variability of our data and in order to ensure that the best performance of the *k-factor* was found, seven possibilities were tested from 3 to 9. The objective was to obtain the saturation point with the optimal *k-factor*.

Figure 4 shows the Box plot representation of the precision, recall, F1-score, and accuracy of the method based on the data distribution (mean + *k-factor* … *SD*) with a single rejection on the EEG signal. Regarding precision, medians decreased and dispersions increased as the *k-factor* increased, indicating less precision as constraint decreased. It was worth noting a drastic decrease in the median of 6 *SD*, compared to the last typical value of 5 *SD*. In relation to recall, it was observed how as the *k-factor* increased, medians did too until saturating at 100% from 6 *SD* on. On the contrary, dispersions decreased from 95.65% in 3 *SD* to 4.34% in 9 *SD*. The third statistical variable to evaluate was the F1-score. Up to the third *k-factor* of 5 *SD* medians were observed to increase to its maximum of 91.98% and then decrease again. The inverse pattern was observed with respect to dispersions, achieving the lowest in 5 *SD*. Finally, accuracy showed again the maximum median together with the lowest dispersion at 5 *SD*, arriving to a median value of 90%. Therefore, it is clear that the best performance was achieved with a *k-factor* of 5, setting the threshold at the mean plus 5 *SD*. Both the F1-score and the accuracy showed the best results with the highest median together with the lowest dispersion. 

### 3.4. A Novel Use of the Artifact Rejection Method Based on Signal Distribution: Double Rejection

The previous section has explained how the threshold based on the distribution of data is usually calculated and used with a single rejection on the EEG signal. This study proposes to introduce the novelty of a double rejection. In the first step the more prominent outliers were discarded to deal with other typical EEG artifacts on the second step. Again, the first rejection was with one of the two motion signals, either the accelerometer or the movement band. The second rejection was on the EEG signal, but only from the remaining epochs. This was performed in two different ways: performing the double rejection with the method based on the data distribution or with a combined method. This last one consisted of removing the movement outliers on the motion signals with the energy-based method and a *k*-value of 99.98% (found previously as optimal in Section 3.2.1) and then, reject the remaining artifacts on the EEG signal with the method based on the data distribution. 

#### 3.4.1. Double Rejection Based on the Data Distribution

In order to perform the double rejection only with the distribution-based method, all *k-factors* tested in Section 3.3.1 were tested again, but using each of them twice; first on the motion signal and second on the remaining EEG signal. 

In order to statistically evaluate this strategy, the analysis was focused on F1-score and accuracy. Precision and recall showed the same performance every time, independent of the method. The higher the *k*-value or *k-factor*, the lower the precision and higher the recall. Therefore, as it was less useful when choosing the optimal *k-factor* it is not shown in Figure 5, despite it was considered in the discussion. 

Figure 5 allow us to evaluate F1-score and accuracy performing the first rejection either on the accelerometer signal (blues) or on the band of movement (reds). First, the double rejection performed on the accelerometer and the EEG signal is analyzed. As for the F1-score, it was observed how dispersions decreased as the *k-factor* increases until 6 *SD*; then, it began to rise again in a much smoother way, without reaching the dispersion values of 3 and 4 *SD*s. Regarding medians, they tended to rise until reaching the maximum at 7 *SD*, then dropped again to 8 and 9 *SD*s. Concerning accuracy, medians increased as *k-factors* did until reaching the maximum at 7 *SD*, from which it began to decrease again. Dispersions decreased as *k-factors* did until reaching the minimum at 5 *SD*, where it began to rise again. Based on the compromise criterion between median and dispersion, the best performance was achieved with a double rejection with a *k-factor* of 6 *SD*. The median values were 91.32% and 89.16% for F1-score and accuracy. Regarding precision and recall, the median values were situated at 96.03% and 90.47%, respectively. 

Second, the double rejection performed on the band of movement and the EEG signal is evaluated. It could be observed how the F1-score improved with each *k-factor* until reaching the lowest dispersion and maximum median at 7 *SD*. The same behavior happened with the accuracy. Therefore, the best performance was achieved at 7 *SD*, with a F1-score median value of 91.02% and an accuracy median value of 89.16%. Regarding precision, the median value was 92.51% and the recall one was 95.32%. 

#### 3.4.2. Double Rejection Based on the Combined Method

The first rejection on the motion signal was performed with the energy-based method and a *k*-value of 99.98%. The second rejection, performed on the remaining EEG epochs, was with the method based on data distribution, testing from 3*SD* to 9*SD*.

The results of the double rejection initiated on the accelerometer signal are shown in blue in Figure 6. Regarding the F1-score, it was observed that the medians increased until 7 *SD* where the maximum was found and then, remained almost constant for the following two *k-factors*. The change in trend of the dispersions appeared earlier. They decreased until 5 *SD*, where it began to increase again until 9 *SD*. In the case of accuracy, the maximum median was achieved at 5 *SD*, from then on it remained almost constant around 90%. Dispersion also improved by being reduced until 5 *SD*, where it began to increase again. Considering both variables and based on the compromise between median and dispersion, the best performance was achieved with 6 *SD*, with medians of 93.05% and 90% for F1-score and accuracy. The median value for precision was 95.29% and for recall, 92.34%. 

In red, Figure 6 shows the double rejection initiated on the band of movement. For both F1-score and accuracy, the maximum median and lowest dispersion were found at 8 *SD*, showing the best performance with median values of 93.32% and 91.67% respectively. Regarding precision, the median was situated at 95.71% and the recall at 91.34%. 

Again, it was observed how the required *k-factor* when performing the first rejection on the band of movement was higher than doing it with the accelerometer, 8 *SD* and 6 *SD*, respectively. 

## 4. Discussion

The complexity and variability of symptoms in patients with RTT required an equally complex and diverse data analysis. The fact of having a database with a wide age range implied a challenge when studying a degenerative disease. The evolution of the disease is divided into stages, where the affectations and capacities change or worsen as the illness evolves [9]. In all stages, the motor ability of the patient is affected; progressively worsening toward an almost complete loss of muscle tone, limb control, or body posture maintenance [24]. In addition to the motor characteristics of each stage, stereotypies, muscle spasms, seizures, and twitches are added to the list of events that directly affect the signal. Depending on the stage and the clinical picture of each patient, their behavior during the session varied, providing different signal patterns. This is shown in Section 3.1, confirming the existence of the three expected groups of registries already found in [25]. Two threshold-based artifact rejection methods have been studied to achieve the best strategy for the three classified groups. Achieving a good performance regardless of the level of artifact indicated the level of robustness and effectiveness of the chosen strategy. In order to evaluate the results, Figure 7 summarizes all the strategies with the *k*-values and *k-factors* chosen as optimal in each case.

The F1-score measures the compromise between precision and recall. For the LmLsd (Low mean Low *SD*) the two methods with the best performance were: the classical method based on data distribution (*gray*) and the hybrid method between the motion band and the EEG (*beige*). However, in databases with low artifact EEGs (common records), this latter method could be time consuming without providing a substantial advantage. The second group was MmMsd (Mid mean Mid *SD*). The most robust method for this type of signals was the hybrid between the band of movement and the EEG (*beige*), although the hybrid method between the accelerometer and the EEG (*dark blue*), the energy-based method with the accelerometer and the EEG (*light blue),* and the classic method with a single rejection (*gray*), would also have a good performance. The third and last group was HmHsd (High mean High *SD*). It is the one that presented the main problem by having high amplitude outliers, which could negatively influence when calculating the thresholds. The method that showed the greater median together with the lowest dispersion was the hybrid between the band of movement and the EEG (*beige*). At the same time and as expected, the classical method with a single rejection was the one that had the lowest median value, since it was the most affected by the presence of outliers. 

The second evaluated variable is accuracy, which measures the percentage of successful detections. In this case, there is one method that clearly showed the best performance in all three groups at the same time, since it had the highest medians together with the lowest dispersions. This method was the hybrid between the band and the EEG (*beige*), whose performance was followed by two other strategies that also showed robustness and had a great performance in all three groups. They were the hybrid between the accelerometer and the EEG (*dark blue*) and the energy-based method with the accelerometer and the EEG (*light blue*). The classic method with a single rejection showed the worst performance with the HmHsd group, demonstrating that it was highly affected by the presence of outliers. Both double rejections based on data distribution (*purple and orange*) and the energy-based method with the band of movement and the EEG (*coral*) also showed inequality in the performance of the three groups, but with better results than the classic method with a single rejection (*gray*). 

Considering both the F1-score and the accuracy, the most robust method to treat all types of signals was the hybrid method between the band of movement and the EEG. The use of the energy-base method to reject high amplitude outliers in the motion signal, together with the rejection of ordinary artifacts in the remaining EEG signal with the method based on the data distribution, proved to be the best strategy when working with databases with high variability. Analyzing the strengths of each method and potentiating them in a common strategy, resulted in the optimal solution for all types of EEG signals, regardless of the level of artifacts. The other methods that also showed a good performance were: the hybrid between the accelerometer and the EEG and the energy-based method with the accelerometer and the EEG. Therefore, both the motion band and the accelerometer demonstrated to be equally helpful in detecting and eliminating motion. However, it is logical that the performance was somewhat better in the band-based method when compared to the manual rejection, since other types of artifacts that were also found in the motion band (due to the high correlation with the EEG) were detected and, consequently, had to be rejected too. 

It is also important to highlight the difference between the double rejection with the energy-based method when using the accelerometer or the motion band. As previously mentioned, when the first rejection was performed with the accelerometer, there were still outliers in the EEG signals. This was because the correlation between the EEG and the motion band was 0.94. This means that most of the motion band was correlated with the EEG. Filtering at low frequencies did not completely isolate movement, since other intrinsic factors of the EEG remained present. However, the correlation between the accelerometer (that is not influenced by other factors) and the EEG was 0.76, which corresponds only to movement. The other artifacts not affecting the accelerometer are the ones that still remained in the EEG signal after the first rejection. In relation to this and as expected, if the first rejection was performed on accelerometer, the optimal *k-factor* needed to be lower than when working with the motion band. The greater the presence of artifacts, the greater the amplitude of the signal in the second rejection, which requires a more restrictive threshold. 

On the other hand, it is important to note that a high percentage of the EEG signal was affected by movement, since the correlation between the EEG and the accelerometer remained high. This can also be seen in the correlation between the motion band and the accelerometer, which was almost equal, 0.77. As predicted in the database description when analyzing the symptomatology of RTT patients, movement played an important role in these records.

The presence of outliers in the two signals to be treated, made the performance of the energy-based method work correctly in both occasions, given the adaptability and robustness of this method against this type of large amplitude artifacts. However, when the same procedure was applied to the motion band and the EEG the performance was worst. This was because in the first rejection, the majority of outliers were already eliminated with the energy-based method. As a consequence, there was no presence of outliers in the second rejection with the EEG, which negatively affected the performance of the energy-based method in front of the signals without outliers. This leads to the limitation of this method: the energy-based method was not efficient to remove elements of high amplitude that were not far away from the distribution of the data (as predicted in Section 2.4.2). For which, the method based on the data-distribution, proved to be more effective. 

On the other hand, we found the limitation of the method based on the data-distribution. As can be seen in Figure 7, despite performing a double rejection with the method based on the data-distribution (*purple and orange*), adequate performance was not achieved in all groups of subjects. The signals of the HmHsd group continued to be penalized (as in the single rejection), demonstrating the high dependence of this method on the data. As also predicted in Section 2.4.1, if the data present outliers with great amplitude or in great number, the method tended to increase the mean and/or the *SD*, only discarding the most prominent outliers and not all those that need to be removed. 

Finally, to end the limitations, it should be noted that some studies where able to automatize the rejection procedure with ICA-based methods owing to artifact localization in time and space, that is, knowing which electrodes were affected (limited number) and how often the artifact was observed [26,27]. However, in our study, using ICA-based methods to reject motion artifacts was not possible due to the difficulty of isolating movement in one or a few components, being present in the vast majority and making their selection extremely difficult even for experts. The same problem was found in other studies [28,29]. These studies concluded that ICA was not effective in removing transient, non-biological artifacts such as abrupt movements or impedance changes due to electrodes displacements. The objective of this study was to obtain a simple and robust artifact rejection method to be used in the clinical environment, without the need for expert supervision. For this reason, ICA approach was discarded for movement detection. 

## 5. Conclusions and Future Work

This study was able to develop a new artifact rejection strategy to deal with the records from pediatric patients with cognitive impairment. The symptomatology of the disease led to great variability, as well as the introduction of uncontrolled movement to the EEG. This was observed in the classification of records, differentiating three groups of data. The combined use of the energy-based method and the distribution-based method has proven to be the most accurate for all of them. 

High amplitude outliers were rejected from the motion signals with the energy-based method. The efficiency was demonstrated both in the accelerometer and in band of movement, obtained from the low-pass filtered EEG signal between 1 and 10 Hz. Therefore, both motion signals have shown to be comparable in the study and detection of movement present in the EEG signals. Likewise, the non-dependence of the method on the data has proven to be fundamental to obtain a robust method for detecting outliers. 

For this reason, the method based on the data-distribution has not achieved a good performance in signals with high amplitude outliers, making insufficient detection even when a double rejection is applied. On the contrary, it has proven to be effective against smaller amplitude artifacts from diverse sources, which have been found to be poorly detected by the energy-based method due to its proximity to the baseline. As a consequence, the method based on the data distribution has been chosen for the second rejection on the EEG signal, achieving the robust rejection that was intended to be pursued. 

Lastly, it is important to set goals for the future. As previously explained, to provide a robust and automatic ICA procedure is not possible with EEG signals with high amounts of motion artifacts [28,29]. However, the application of automatic ICA algorithms after removing highly artifact time-segments may improve the signal quality. This issue is out of the scope of this research study, but it may be addressed in future studies. 

## Figures and Tables

**Figure 1 entropy-23-01030-f001:**
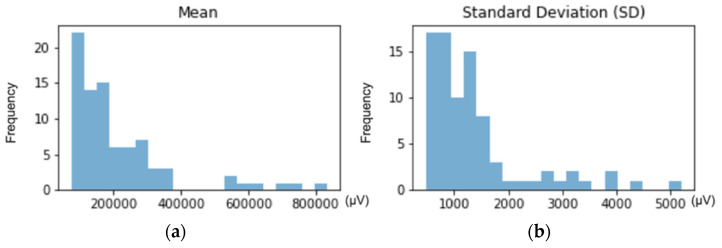
Describes the data distribution from the mean and *SD*: (**a**) Histogram showing the distribution of the means; (**b**) histogram showing the distribution of the *SD*s; (**c**) mean-*SD* ratio distribution using kernel density estimation (KDE); (**d**) density representation as the weighted sum of Gaussian distributions using the Gaussian mixture model (GMM). Three groups were differentiated: 1. LmLsd (blue): low mean and low *SD*. 2. MmMsd (orange): mid mean and mid *SD*. 3. HmHsd (green): high mean and high *SD*.

**Figure 2 entropy-23-01030-f002:**
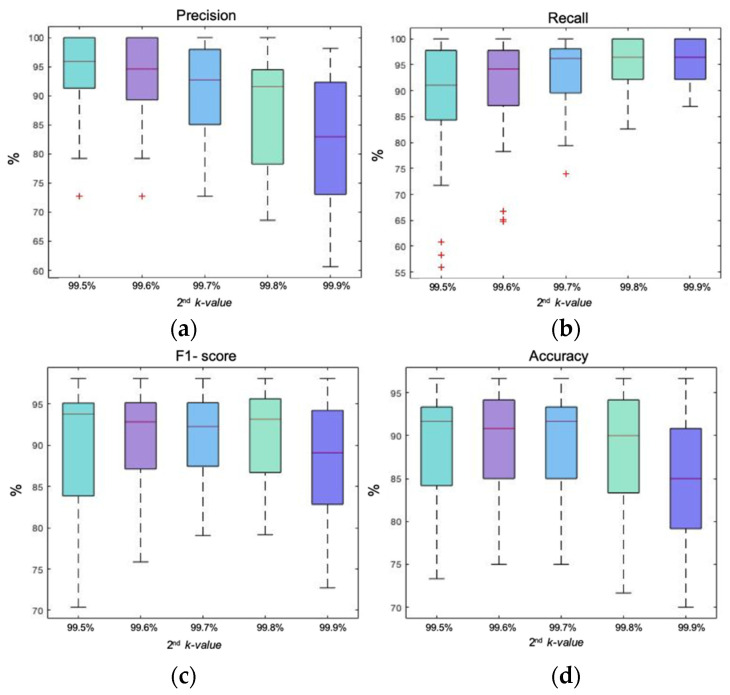
Box plot representation of precision, recall, F1-score, and accuracy of the energy-based method of double rejection. The first rejection was performed with the accelerometer signal with a *k*-value of 99.98%. The second rejection was performed on the remaining epochs of the EEG signal with five different *k*-values (99.5% to 99.9%); (**a**) box plot showing the precision achieved with each *k*-value; (**b**) box plot showing the recall achieved with each *k*-value; (**c**) box plot representing the F1-score achieved with each *k*-value; (**d**) box plot showing the accuracy achieved with each *k*-value.

**Figure 3 entropy-23-01030-f003:**
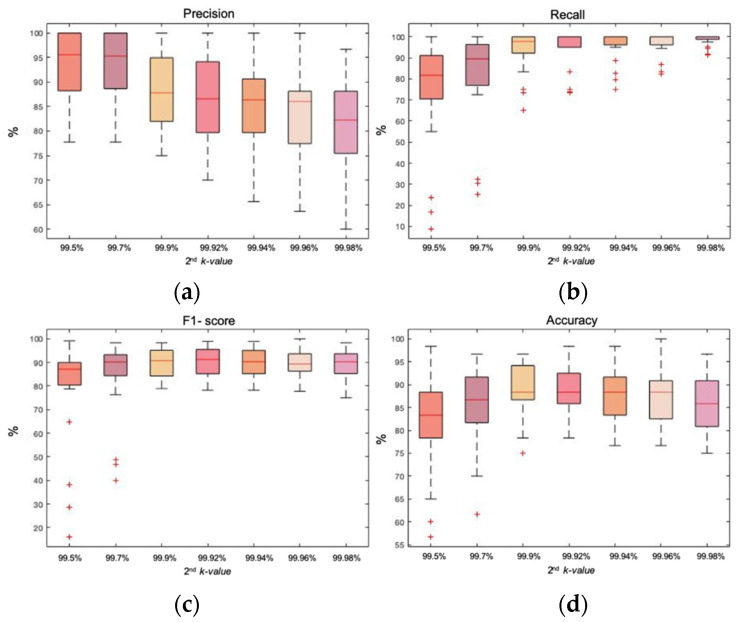
Box plot representation of the precision, recall, F1-score, and accuracy of the energy-based method of double rejection. The first rejection is performed on the band of movement with a *k*-value of 99.98%. The second rejection is performed on the remaining epochs of the EEG signal with five different *k*-values; (**a**) box plot showing the precision achieved with each *k*-value; (**b**) box plot showing the recall achieved with each *k*-value; (**c**) box plot representing the F1-score achieved with each *k*-value; (**d**) box plot showing the accuracy achieved with each *k*-value.

**Figure 4 entropy-23-01030-f004:**
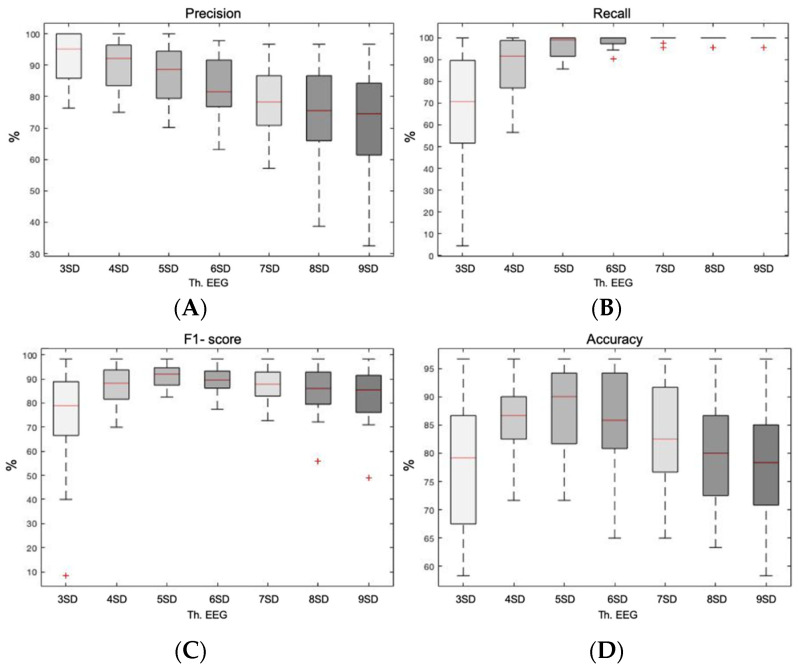
Box plot representation of the precision, recall, F1-score, and accuracy of the method based on the data distribution (mean + *k-factor* × *SD*) with a single rejection on the EEG signal. (**A**) Box plot showing the precision achieved with each *k-factor.* (**B**) Box plot showing the recall achieved with each *k-factor*. (**C**) Box plot representing the F1-score achieved with each *k-factor*. (**D**) Box plot showing the accuracy achieved with each *k-factor*.

**Figure 5 entropy-23-01030-f005:**
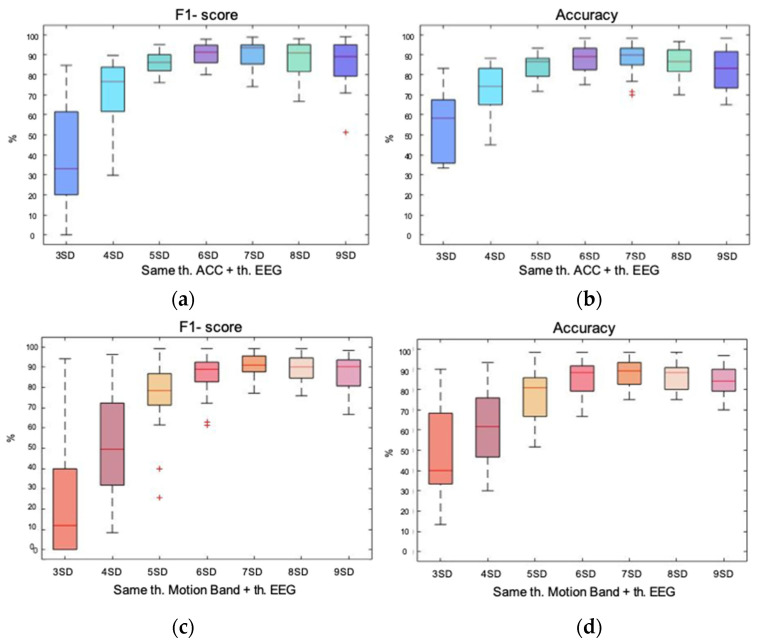
Box plot representation of the F1-score and accuracy of the data distribution-based method of double rejection. In blues, the first rejection is performed on the accelerometer signal and in reds, the first rejection is performed on the signal of the band of movement. The second rejection is performed on the remaining epochs of the EEG signal of each first rejection. In both rejections same *k-factors* are applied, all repeated twice from 3 *SD* to 9 *SD*; (**a**) box plot showing the F1-score achieved with each *k-factor* for the double rejection performed on the accelerometer and the EEG; (**b**) box plot showing the accuracy achieved with each *k-factor* for the double rejection performed on the accelerometer and the EEG; (**c**) box plot showing the F1-score achieved with each *k-factor* for the double rejection performed on the band of movement and the EEG; (**d**) box plot showing the accuracy achieved with each *k-factor* for the double rejection performed on the band of movement and the EEG.

**Figure 6 entropy-23-01030-f006:**
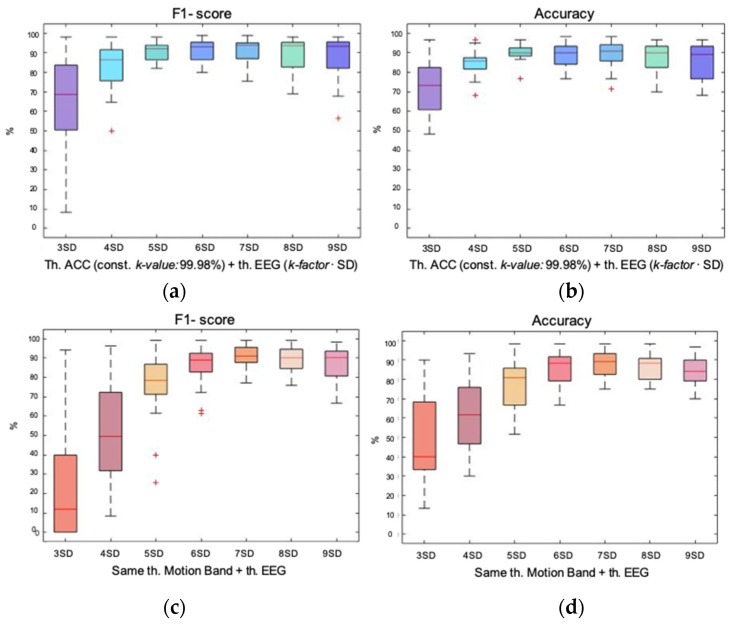
Box plot representation of the F1-score and accuracy of the hybrid method of double rejection. In blues, the first rejection is performed with the energy-based method and a constant *k*-value of 99.98% on the accelerometer signal; and in reds, the first rejection is performed equally but on the signal of the band of movement. The second rejection is performed on the remaining epochs of the EEG signal of each first rejection with the distribution-based method. Seven *k-factors* were tested from 3 *SD* to 9 *SD*; (**a**) box plot showing the F1-score achieved with each *k-factor* for the double rejection performed on the accelerometer and the EEG; (**b**) box plot showing the accuracy achieved with each *k-factor* for the double rejection performed on the accelerometer and the EEG; (**c**) box plot showing the F1-score achieved with each *k-factor* for the double rejection performed on the band of movement and the EEG; (**d**) box plot showing the accuracy achieved with each *k-factor* for the double rejection performed on the band of movement and the EEG.

**Figure 7 entropy-23-01030-f007:**
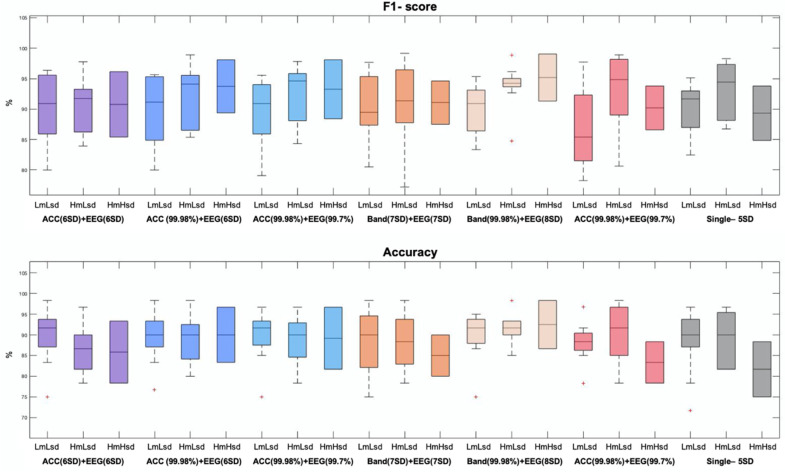
Box plot showing the results of the F1-score and the accuracy for the chosen *k*-values and *k-factor*s in each of the strategies evaluated in the result section. Each performance is divided between the three groups of signals: LmLsd, MmMsd, and HmHsd (described in Figure 1d).

**Table 1 entropy-23-01030-t001:** Demographic characteristics of participants (n = 29).

Variable	Mean ± SD	Range
Age (years)	10.25 ± 4.73	(3–19)
Weight (kg)	28.81 ± 14.46	(12–52)
Head circumference (cm)	48.25 ± 1.53	(45–50)
Regression age (months)	14.93 ± 7.56	(7–36)
Rett stage	3.13 ± 0.74	(2–4)
Walking	0.44 ± 0.51	(0–1)
Ability to walk	2.00 ± 1.04	(1–3)
Speech	2.73 ± 0.46	(2–3)
Hand use	2.07 ± 0.96	(1–3)
Scoliosis	1.44 ± 1.41	(0–3)

Walking: 0 = no walking; 1 = unsupported walking; 2 = walking with support. Ability to walk: 0 = Normal gait; 1 = Mildly apraxic; 2 = Severely apraxic or requiring to be held. Speech: 0 = Normal; 1 = Sentences/phrases; 2 = Single words; 3 = Non-verbal. Hand use: 0 = Normal; 1 = Purposeful grasping; 2 = Tapping for needs; 3 = No hand use. Scoliosis: 0 = Not present; 1 = <20°; 2 = <30°; 3 = >30° or if surgical correction had taken place.

## Data Availability

The datasets presented in this article are not readily available because the informed consent did not include the declaration regarding publicity of clinical data. Requests to access the datasets should be directed to Hospital Sant Joan de Déu (https://www.sjdhospitalbarcelona.org/ (accessed on 11 August 2021)).

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
