# Peer review of "Choosing Strategies to Deal with Artifactual EEG Data in Children with Cognitive Impairment"

_entropy, 2021, doi:10.3390/e23081030_

Round 1

Reviewer 1 Report

The paper presents an interest application of EEG artidacts suppression/reduction in specific neurodegenerative diseases in children. The specific conditions imposes constraints on EEG extraction for diverse kind of movements (i.e., stereotyped) during acquisition.

It is not so clear to me why in this case some procedures based on ICA and more than second order moments of the estimated distribution or else in entropic content (i.e., permutation entropy, Tsallis entropy) are not appropriate.

The paper is well written and organized, perhaps some more info can be added on ethics aspects taking into account that data are taken from minors. 

I suggest to develop a paragraph on ICA approaches to artifacts and the possibility of using wavelet analysis and wICA approaches. Please refer to the following works:

Mammone, N.; Morabito, F.C. Enhanced Automatic Wavelet Independent Component Analysis for Electroencephalographic Artifact Removal. Entropy 201416, 6553-6572. 

Castellanos N., Makarov V., Recovering EEG brain signals: artifact suppression with wavelet enhanced independent component analysis, J. of Neuroscience Methods, 2006 Dec 15;158(2):300-12. doi: 10.1016/j.jneumeth.2006.05.033. Epub 2006 Jul 7.

Author Response

We appreciate your positive review. We have thoroughly revised the manuscript to address your concerns and suggestions. The feedback provided has enabled us to improve the manuscript. The answer to the review is in the attached file (please see the attachment). 

Reviewer 2 Report

The manuscript is too wordy in almost every section and paragraph, and the authors should keep in mind that a manuscript must be written comprehensively without repetitions and redundancies.

Although artifact detection applies to a wide range of applications, the presented research only concerns one specific context, i.e., Rett Syndrome (RTT), resulting in a very limited number of cases. The author collected 29 cases in three categories on stages, with no details on the classification or the number of each class. There are numerous ways to calculate the entropy of a signal or a distribution. None of them is cited when the authors refer to their method. It is not clear in the manuscript methods how the entropy is calculated from the signal. Nevertheless, there are indications of few if any novelties in the methods.

Although the language used is not entirely bad, the text is too wordy, confusing, and imprecise sometimes. There are several commas misplaced or missing, abuse of passive voice contributing to text imprecisions. There is a broken reference at line 164.

The main drawbacks of the research are the lack of novelty in methods and the very limited data representativity. Therefore, this reviewer believes it is complicated, if possible, to overcome these drawbacks.

Author Response

(The authors gave the same response as above.)

Round 2

Reviewer 2 Report

The authors addressed each of the raised points adequately, modifying the manuscript to fix most raised issues.  Text writing is pretty acceptable. Although the main drawback of the manuscript is still the weak novelty, this reviewer accepts the argument of the particular application context.